# Chronic Alcohol Consumption is Inversely Associated with Insulin Resistance and Fatty Liver in Japanese Males

**DOI:** 10.3390/nu12041036

**Published:** 2020-04-09

**Authors:** Takemi Akahane, Tadashi Namisaki, Kosuke Kaji, Kei Moriya, Hideto Kawaratani, Hiroaki Takaya, Yasuhiko Sawada, Naotaka Shimozato, Yukihisa Fujinaga, Masanori Furukawa, Koh Kitagawa, Takahiro Ozutsumi, Yuuki Tsuji, Daisuke Kaya, Hiroyuki Ogawa, Hirotetsu Takagi, Koji Ishida, Hitoshi Yoshiji

**Affiliations:** Department of Gastroenterology, Nara Medical University, Shijo-cho 840, Kashihara, Nara 634-8522, Japan; tadashin@naramed-u.ac.jp (T.N.); kajik@naramed-u.ac.jp (K.K.); moriyak@naramed-u.ac.jp (K.M.); kawara@naramed-u.ac.jp (H.K.); htky@naramed-u.ac.jp (H.T.); yasuhiko@naramed-u.ac.jp (Y.S.); shimozato@naramed-u.ac.jp (N.S.); fujinaga@naramed-u.ac.jp (Y.F.); furukawa@naramed-u.ac.jp (M.F.); kitagawa@naramed-u.ac.jp (K.K.); ozutaka@naramed-u.ac.jp (T.O.); tsujih@naramed-u.ac.jp (Y.T.); kayad@naramed-u.ac.jp (D.K.); ogawah@naramed-u.ac.jp (H.O.); htakagi@naramed-u.ac.jp (H.T.); ishidak@naramed-u.ac.jp (K.I.); yoshijih@naramed-u.ac.jp (H.Y.)

**Keywords:** alcohol consumption, insulin resistance, fatty liver

## Abstract

We aimed to elucidate the effect of chronic alcohol consumption on fatty liver. We assessed the consumption of alcohol in 2429 Japanese males (mean age: 54.2 ± 9 years); they were classified according to average consumption into non-drinkers (ND), light drinkers (LD), moderate drinkers (MD), and heavy drinkers (HD). The prevalence of fatty liver was the lowest in the MD and highest in the ND group (*p* < 0.001), while obesity was not significantly different among the groups (*p* = 0.133). Elevated levels of alanine aminotransferase (ALT) were the lowest in the MD group (*p* = 0.011) along with resistance to insulin (homeostasis model assessment-insulin resistance (HOMA-IR)), which was highest in the ND group (*p* = 0.001). Chronic consumption of alcohol was independently and inversely associated with fatty liver and insulin resistance after adjusting for obesity, hypertension, fasting hyperglycemia, habit of drinking sweet beverages, physical activity, and age (odds ratios are as follows: ND, 1; LD, 0.682; MD, 0.771; HD, 0.840 and ND, 1; LD, 0.724; MD, 0.701; HD, 0.800, respectively). We found that regardless of the type of alcoholic beverage, chronic consumption of alcohol is inversely associated with insulin resistance and fatty liver in Japanese males. This study had limitations, most notably the lack of investigation into diet and nutrition.

## 1. Introduction

Alcohol liver disease (ALD) is a major cause of liver disease worldwide. Excess alcohol consumption is reported to lead to ALD, and 90% percent of people consuming more than 60 g of alcohol per day have been shown to develop steatosis [1]. Non-alcoholic fatty liver disease (NAFLD) is another etiological agent of hepatic steatosis. It should be noted that NAFLD has been strongly associated with obesity and has been considered a phenotype of the metabolic syndrome (Mets) in the liver [2,3,4,5]. In addition, resistance to insulin has been implicated in the pathogenesis of NAFLD [5]. Recent studies have reported that light to moderate consumption of alcohol may be associated with lower cardiovascular mortality [6,7,8] and a reduced risk of developing type 2 diabetes [9,10,11]. Other reports have demonstrated that light to moderate consumption of alcohol might protect against fatty liver [12,13]. Likewise, modest consumption of alcohol was associated with a decreased risk of incident hepatic steatosis [14]. In contrast, a study showed that moderate consumption of alcohol was positively associated with the risk of NAFLD in elderly Chinese men [15]. On the other hand, a recent report demonstrated that there is insufficient evidence for or against the role of moderate consumption of alcohol in the development of NAFLD [16]. Therefore, the influence of light to moderate consumption of alcohol on the liver remains elusive.

In this study, we investigated the relationship between the consumption of alcohol and the prevalence of fatty liver and resistance to insulin in Japanese men. 

## 2. Materials and Methods 

### 2.1. Study Population

Our cross-sectional study used data from 2461 males (age range: 27–89 years) who had consecutively undergone comprehensive annual examinations for employee or resident health check-ups at Nara Health Promotion Center during 2012. Most subjects were asymptomatic. Of these, 32 were excluded from the study for being positive for the hepatitis B surface antigen or anti-hepatitis C virus antibody. The remaining 2429 participants (mean age: 54.2 ± 9 years, age range: 27–89 years) were included in this study. This study was approved by the Ethics Committee of Nara Medical University.

### 2.2. Clinical and Laboratory Assessments

The information collected using a standardized questionnaire included alcohol consumption, habits of drinking sweet beverages and eating greasy meat, and physical activity (Appendix A). The body mass index (BMI) was calculated as weight (kg)/squared height (m^2^). Obesity was defined as a BMI of ≥25 kg/m^2^ according to the criteria established by the Japan Society of Obesity. Waist circumference (WC) was measured at the level of the umbilicus. All subjects were examined following a 12-h overnight fast. Sitting blood pressure was measured using an automatic sphygmomanometer. Hypertension was defined as a systolic blood pressure (SBP) of ≥140 mmHg, a diastolic blood pressure (DBP) of ≥90 mmHg, or current use of antihypertensive medications.

All biochemical analyses were performed using standard techniques. Fasting hyperglycemia was defined as a fasting blood glucose of ≥110 mg/dL or current use of antidiabetic medications. An elevated level of alanine aminotransferase (ALT) was defined as ≥ 30 IU/L. Resistance to insulin was assessed using the homeostasis model assessment-insulin resistance (HOMA-IR), which was calculated as the fasting insulin (μU/mL) × the fasting glucose (mg/dL)/405, and was defined as ≥2.5. According to the Japanese criteria, metabolic syndrome was defined as a WC of ≥85 cm and the presence of two or more of the following risk factors: medicated for hypertension, SBP of ≥130 mmHg or DBP of ≥85 mmHg; medicated for dyslipidemia or high-density lipoprotein (HDL) cholesterol of <40 mg/dL and/or triglycerides of ≥150 mg/dL; and medicated for diabetes or fasting blood glucose of ≥110 mg/dL.

Fatty liver was identified by ultrasonography when increased liver echogenicity (“bright liver”) and a contrast between the hepatic and renal parenchymal tissues (“liver-kidney contrast”) were recognized. Ultrasonography was performed using the LOGIQ 7, with a 4-MHz convex array transducer (GE Healthcare, Waukesha, WI, USA).

### 2.3. Evaluation of Alcohol Consumption

Each subject reported their frequency of consumption of alcohol per week, as well as the types and volume of alcohol consumed per day for the last year. The consumption of alcohol in grams was calculated based on the information provided, by using the representative percent of alcohol for each type of alcoholic drink: 5% for beer, 15% for sake, 25% for shochu, 40% for whiskey, and 12% for wine. To calculate the average consumption of alcohol per day in grams of ethanol, the frequency of the consumption of alcohol per week was multiplied by the grams of ethanol per day and divided by 7 days. Subjects were then classified according to their consumption of alcohol: non-drinker (ND), 0 g/d; light drinker (LD), <30 g/d; moderate drinker (MD), 30–60 g/d; and heavy drinker (HD), ≥60 g/d.

### 2.4. Statistical Analysis

Data were expressed as the mean ± standard deviation. Continuous variables were compared using a one-way analysis of variance, and the significance of individual differences was evaluated by using the Bonferroni test if the ANOVA was significant. Categorical variables were compared using the chi-square test.

Multivariate logistic regression analyses were used to assess whether the categories of consumption of alcohol were independently associated with the prevalence of fatty liver and resistance to insulin. A *p*-value of <0.05 was considered to be statistically significant. All calculations were performed using the SPSS software (Version 25, Inc., Chicago, IL, USA).

## 3. Results

### 3.1. Clinical Characteristics 

The characteristics of the study population are shown in Table 1. There were 894 (36.8%) ND, 857 (35.3%) LD, 444 (18.3%) MD, and 234 (9.6%) HD. The observed BMI was not significantly different among the four groups. The percentage of those who often drink sweet beverages was significantly decreased in association with the consumption of alcohol. The adjusted standardized residuals in the ND, LD, MD, and HD groups were 5.8, −0.1, −3.8 and −4.4, respectively. The percentage of those who often eat greasy meat was not significantly different among the four groups. The percentage of physical activity was higher in the LD and MD groups than those in the ND and HD groups. The adjusted standardized residuals in the ND, LD, MD, and HD groups were −4.2, 2.8, 1.0 and 0.9, respectively. The SBP and DBP, prevalence of hypertension, and the percentage of the use of antihypertensive medications were shown to be significantly increased in association with the consumption of alcohol (all *p* < 0.001). Similarly, the levels of aspartate aminotransaminase (AST), gamma-glutamyl transferase (GGT), and triglycerides were also significantly increased across the categories with an increasing consumption of alcohol (all *p* < 0.001). However, we observed a U-shaped association between the level of ALT and the categories of consumption of alcohol. Respectively, HOMA-IR was noted to be significantly higher in the ND group than those in the LD, MD, and HD groups. The prevalence of metabolic syndrome was noted to be the lowest in the LD group and highest in the HD group. The adjusted standardized residuals in the ND, LD, MD, and HD groups were −0.6, −2.8, 1.0 and 4.2, respectively. The percentages of the use of lipid-lowering medications or antidiabetic medications were not significantly different among the four groups.

### 3.2. Comparisons of Prevalence of Fatty Liver, Obesity, Elevated Levels of ALT and Resistance to Insulin among the 4 Groups 

The prevalence of fatty liver was shown to be the lowest in the MD group and highest in the ND group (*p* < 0.001). There was a U-shaped association observed between the prevalence of fatty liver and the categories of consumption of alcohol. The adjusted standardized residuals in the ND, LD, MD, and HD groups were 4.2, −2.8, −1.9 and 0.2, respectively (Figure 1A). However, the prevalence of obesity (body mass index of ≥ 25 kg/m^2^) was not significantly different among the four groups (*p* = 0.133) (Figure 1B). The prevalence of elevated levels of ALT (>30 IU/L) was exhibited to be the lowest in the MD group (*p* = 0.011). The adjusted standardized residuals in the ND, LD, MD, and HD groups were 2.6, −1.8, −2.0 and 1.2, respectively. Similarly, we noted a U-shaped association between the prevalence of elevated levels of ALT and the categories of consumption of alcohol (Figure 1C). The prevalence of resistance to insulin (HOMA-IR ≥ 2.5) was demonstrated to be the lowest in the MD group and highest in the ND group (*p* = 0.001). The adjusted standardized residuals in the ND, LD, MD, and HD groups were 3.9, −1.9, −2.4 and −0.2, respectively. Accordingly, there was a U-shaped association shown between the prevalence of the resistance to insulin and the categories of consumption of alcohol (Figure 1D).

### 3.3. Predictors of Fatty Liver and Resistance to Insulin

Chronic consumption of alcohol was demonstrated to be independently and inversely associated with fatty liver after adjusting for age, obesity, hypertension, fasting hyperglycemia, habit of drinking sweet beverages, and physical activity. The odds ratios were shown to be as follows: ND, 1; LD, 0.682; MD, 0.771; and HD, 0.840. Light consumption of alcohol exhibited the most decreased odds ratio (Table 2). Similarly, chronic consumption of alcohol was observed to be independently and inversely associated with resistance to insulin after adjusting for age, obesity, hypertension, fasting hyperglycemia, habit of drinking sweet beverages, and physical activity. The odds ratios were shown to be as follows: ND, 1; LD, 0.724; MD, 0.701; and HD, 0.800. In this case, moderate consumption of alcohol was noted to exhibit the most decreased odds ratio (Table 3).

### 3.4. Comparisons of Biological Features among Non-Drinkers and Various Types of Drinkers

Consumption of alcohol and the percentage of the HD group were demonstrated to be the highest in mixed drinkers (54.3 ± 23.9 g/d). The BMI was not shown to be significantly different among the ND group and various types of drinkers. Similarly, the level of ALT was not observed to be significantly different among the ND group and various types of drinkers; however, ALT tended to be the highest in the ND group. On the other hand, the level of GGT was observed to be the lowest, whereas HOMA-IR and the prevalence of fatty liver were reported to be the highest in the ND group. The percentage of the use of antidiabetic medications was not significantly different among the ND group and various types of drinkers; however, it tended to be higher in the ND group (*n* = 62, 6.9%) when compared to all drinkers (*n* = 89, 5.8%) (Table 4). The prevalence of fatty liver was not shown to be significantly different among the various types of drinkers in the LD, MD, or HD groups (Table 5).

## 4. Discussion

This study showed that consumption of alcohol is inversely associated with insulin resistance and fatty liver. We observed a U-shaped relationship between the consumption of alcohol and the prevalence of fatty liver and elevated levels of ALT. Similarly, we observed the same relationship between the consumption of alcohol and the prevalence of elevated resistance to insulin. The prevalence of fatty liver and resistance to insulin was shown to be the lowest in the MD group, whereas they were observed to be the highest in the ND group. The percentage of those who often drink sweet beverages was significantly decreased in association with alcohol consumption, and the percentage of physical activity was significantly higher in the LD and MD groups than those in the ND and HD groups. To avoid potential cofounders, we performed a multivariate logistic analysis. We demonstrated that chronic consumption of alcohol was independently and inversely associated with fatty liver and resistance to insulin after adjusting for obesity, hypertension, fasting hyperglycemia, habit of drinking sweet beverages, physical activity, and age. Moreover, regardless of the type of the alcoholic beverages, HOMA-IR and the prevalence of fatty liver was shown to be lower in the various types of drinkers compared with the ND group. 

Excessive consumption of alcohol is known to be harmful and has been associated with cancer, cardiovascular diseases, type 2 diabetes, liver cirrhosis, and stroke [17,18]. In contrast, moderate consumption of alcohol has been shown to exert beneficial effects on coronary heart disease, stroke, and type 2 diabetes mellitus [19,20,21]. However, this mostly pertains to what extent moderate consumption of alcohol might be cardioprotective, illustrated by the so-called ‘J-’shaped curve exhibited by the relationship between the consumption of alcohol and overall mortality [22]. In this study, chronic consumption of alcohol was observed to significantly reduce the prevalence of fatty liver after adjusting for obesity, hypertension, fasting hyperglycemia, habit of drinking sweet beverages, physical activity, and age. Concordant with our results, Knott et al. reported a reduced risk of type 2 diabetes in all levels of alcohol intake <63 g/d [23]. Sookoian et al. also showed that the protective effect of light to moderate consumption of alcohol on NAFLD was independent of covariates, such as BMI [24]. In the meta-analysis of heavy drinkers vs. non-drinkers, including 6 Japanese and 2 German studies, heavy consumption of alcohol was not statistically associated with the risk of fatty liver. However, in further subgroup analysis, the heavy consumption of alcohol was associated with a 33.7% reduction in the risk of fatty liver in Japanese men [25]. Approximately 40–50% of the Japanese population are known to carry an inactive ALDH2 gene (ALDH2 2*2 allele). Due to the accumulation of acetaldehyde, people carrying the ALDH2*2 allele are thought to experience heightened responses to alcohol, thus leading to lower rates of drinking. In this study, the average consumption of alcohol in the HD group was 77.4 g/d (95% CI: 75.0, 79.8). Therefore, the direct toxic effects of alcohol in this study might be less than those reported in European studies.

Some reports suggest that moderate consumption of alcohol has a protective effect against the development of resistance to insulin [26,27]. Our results showed that chronic consumption of alcohol, even in the HD group, was able to confer significant protection from resistance to insulin. Although the mechanisms of the beneficial effects of chronic consumption of alcohol on fatty liver are not clear, one of these effects might be explained by increased sensitivity to insulin. Other studies have reported that chronic consumption of alcohol improved the sensitivity to insulin via increased levels of hepatic glutathione, which is known to increase the hepatic insulin sensitizing substance [28,29]. An in vivo study has suggested that a moderate intake of alcohol diminished the development of NAFLD through the sirtuin-1/-adiponectin-dependent signaling cascades [30]. In contrast, another in vivo study showed that binge drinking induced systemic resistance to insulin [31]. Our classification of the consumption of alcohol did not reflect binge drinking, as such binge drinking and chronic consumption of alcohol might have different effects on the sensitivity to insulin. The prevalence of metabolic syndrome was the highest in the HD group among the four categories of alcohol consumption. The reason was that triglycerides and blood pressure, the components of metabolic syndrome, were the highest in the HD group as alcohol intake is associated with triglycerides and blood pressure in a dose-dependent manner [32].

In the present study, GGT and AST were increased across the categories of alcohol consumption, whereas there was a U-shaped association between the level of ALT and the categories of alcohol consumption. GGT, ALT, and AST are the most frequently used markers for the early detection of alcoholic liver disease (ALD). GGT is usually higher in patients with ALD compared to those who have other liver diseases. An elevation of AST is observed in ALD, and ALT levels are commonly lower than AST. However, all these laboratory values are not specific for ALD, and they have low sensitivity and specificity. ALT has the lowest sensitivity among them [33]. The level of these values is also often elevated in NAFLD patients, and ALT levels are commonly higher than AST. Therefore, the elevation of ALT due to NAFLD in the ND group, elevation of ALT due to ALD in the HD group, and the lower prevalence of fatty liver in the LD and MD groups were observed, resulting in the U-shaped relationship between the level of ALT and the categories of alcohol consumption. 

It has been reported that modest consumption of wine is associated with a reduced prevalence of NAFLD [34]. However, there was no relationship observed between types of alcoholic beverages and fatty liver in this study. In the present study, the number of wine drinkers was quite small because the habit of daily consumption of wine is uncommon among the Japanese. Instead, beer, sake, and shochu are preferentially consumed in Japan. The prevalence of fatty liver was demonstrated to be significantly lower in any type of alcohol drinkers relative to non-drinkers. In addition, HOMA-IR was observed to be significantly lower in any type of alcohol drinkers compared with that of non-drinkers. Therefore, regardless of the type of alcoholic beverage, it was suggested that chronic consumption of alcohol was inversely associated with resistance to insulin and fatty liver.

Our study had some limitations. First, we did not enroll female subjects. Although 2146 female subjects were investigated, 1620 (75%) subjects did not drink, and the number of those who drank more than 20 g/d was only 172, while the number of subjects with fatty liver was only 24 among those subjects. Therefore, the sample size was insufficient to elucidate the association between chronic consumption of alcohol and fatty liver in women by the methods used in the present study. Second, this study was a cross-sectional study. Third, as hepatic steatosis was diagnosed by ultrasonography, there was limited accuracy for the detection of mild steatosis. The assessments of liver were performed by ultrasound which bears limitations such as the detection of hepatitis and fibrosis. Fourth, most of the subjects were asymptomatic and able to engage in their work. The results of this study might, therefore, not be applicable to individuals who have liver fibrosis or other liver diseases. Fifth, the subjects only provided their habit of drinking sweet beverages and eating greasy meat. We did not further investigate their diet or nutrition, especially fructose and saturated fatty acids, which have been shown to play a pivotal role in the development of fatty liver. Sixth, the study was conducted on the normal-weight Japanese population, which is very different from that of western countries, where fatty liver occurs in overweight or obese individuals with insulin resistance and often with diabetes, and where the most prevalent consumption of alcohol is represented by wine. Seventh, the subjects only provided their consumption of alcohol for the last year. We did not investigate former drinkers and lifetime drinking history.

## 5. Conclusions

In conclusion, chronic consumption of alcohol is inversely associated with insulin resistance and fatty liver in Japanese males. However, the results demonstrated in the present study do not suggest the safeness of alcohol consumption for everyone. Thus, the threshold for harm induced by the consumption of alcohol remains unclear.

## Figures and Tables

**Figure 1 nutrients-12-01036-f001:**
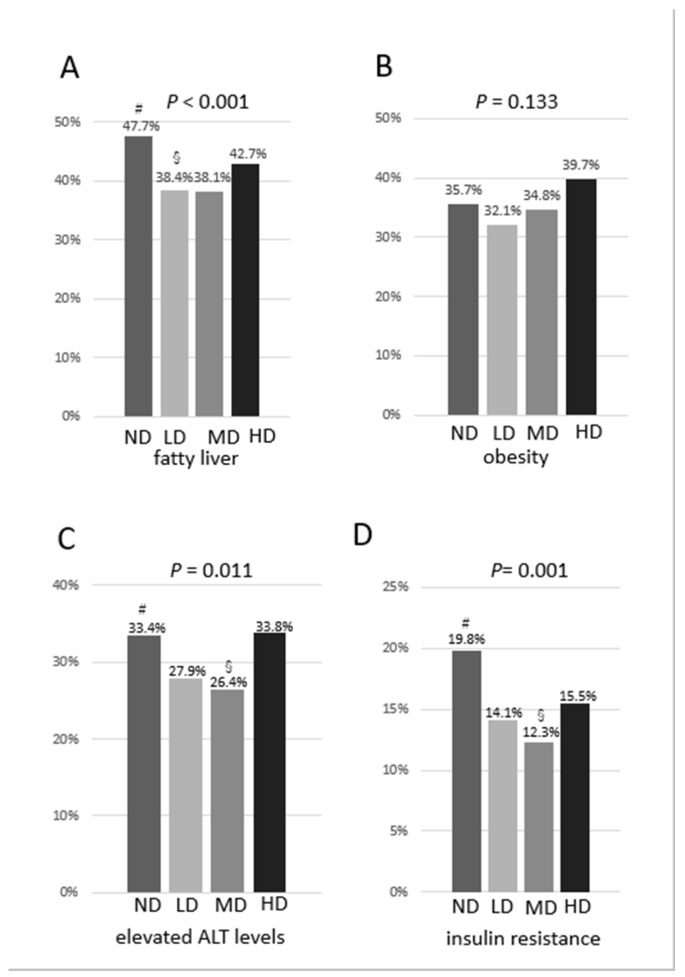
(**A**) Prevalence of fatty liver by categories of consumption of alcohol; (**B**) Prevalence of obesity by categories of consumption of alcohol; (**C**) Prevalence of elevated levels of ALT by categories of consumption of alcohol; (**D**) Prevalence of resistance to insulin by categories of consumption of alcohol. #adjusted standardized residual > 1.96; § adjusted standardized residual < −1.96.

**Table 1 nutrients-12-01036-t001:** Clinical characteristics according to categories of consumption of alcohol.

	ND	LD	MD	HD	*p*-Value
N (%)	894 (36.8)	857 (35.3)	444 (18.3)	234 (9.6)	
Alcohol consumption (95% CI) (g/d)	0 (0, 0)	15.8 ± 5.9 (15.4, 16.2)	41.6 ± 5.8 (41.0, 42.1)	77.4 ± 18.8 (75.0, 79.8)	<0.001
Age (y)	53.4 ± 9.2	54.7 ± 9.8	54.7 ± 8.5	54.3 ± 8.5	0.016
BMI (kg/m^2^)	24.2 ± 3.4	23.9 ± 3.1	24.0 ± 3.0	24.4 ± 3.0	0.074
Habit of drinking sweet beverages (%)	322 (36) ^#^	248 (28.9)	96 (21.6) ^§^	39 (16.7) ^§^	<0.001
Habit of eating greasy meat (%)	322 (36)	315 (36.8)	184 (41.4)	96 (41)	0.162
Physical activity (%)	316 (35.3) ^§^	366 (42.7) ^#^	187 (42.1)	84 (35.9)	0.005
Systolic BP	127 ± 15	129 ± 14	131 ± 15 ***	134 ± 14 ***	<0.001
Diastolic BP	78 ± 11	79 ± 11	82 ± 10 ***	84 ± 11 ***	<0.001
Hypertension (%)	277 (31) ^§^	304 (35.5)	191 (43) ^#^	137 (58.5) ^#^	<0.001
Use of antihypertensive medications	165 (18.5) ^§^	173 (20.2)	122 (27.5) ^#^	72 (30.8) ^#^	<0.001
AST (IU/L)	24 ± 19	24 ± 7	26 ± 22 **	28 ± 14 ***	<0.001
ALT (IU/L)	30 ± 19	27 ± 15 **	27 ± 20	30 ± 20	0.001
GGT (IU/L)	34 ± 30	42 ± 36 *	62 ± 53 ***	96 ± 119 ***	<0.001
Triglycerides (mg/dL)	125 ± 86	123 ± 76	139 ± 103 *	169 ± 148 ***	<0.001
Total cholesterol (mg/dL)	205 ± 33	207 ± 32	206 ± 32	207 ± 34	0.524
HDL cholesterol (mg/dL)	53 ± 13	56 ± 13 ***	60 ± 15 ***	61 ± 15 ***	<0.001
LDL cholesterol (mg/dL)	129 ± 29	127 ± 29	120 ± 31 ***	116 ± 33 ***	<0.001
Use of lipid-lowering medications (%)	129 (14.4)	94 (11.0)	53 (11.9)	21 (9.0)	0.054
Fasting glucose (mg/dL)	103 ± 20	103 ± 20	105 ± 20	107 ± 21	0.24
Use of antidiabetic medications (%)	62 (6.9)	47 (5.5)	27 (6.1)	15 (6.4)	0.657
HOMA-IR	1.9 ± 1.7	1.6 ± 1.4 **	1.5 ± 1.0 ***	1.6 ± 1.2 *	<0.001
Metabolic syndrome (%)	202 (22.6)	172 (20.1) ^§^	111 (25.1)	80 (34.2) ^#^	<0.001

Quantitative variables are presented as mean ± standard deviation. CI, confidence interval; BMI, body mass index; BP, blood pressure; AST, aspartate aminotransferase; ALT, alanine aminotransferase; GGT, gamma-glutamyl transferase; HOMA-IR, homeostasis model assessment-insulin resistance. * *p* < 0.05 vs. ND; ** *p* < 0.005 vs. ND; *** *p* < 0.001 vs. ND. ^#^ adjusted standardized residual > 1.96; ^§^ adjusted standardized residual < −1.96.

**Table 2 nutrients-12-01036-t002:** Predictors of fatty liver.

	Odds Ratio (95% CI)	*p*-Value
ND	1	
LD	0.682 (0.549–0.846)	0.001
MD	0.771 (0.675–0.880)	<0.001
HD	0.840 (0.749–0.941)	0.003

Explanatory variables include age, obesity, hypertension, fasting hyperglycemia, habit of drinking sweet beverages, and physical activity. CI, confidence interval.

**Table 3 nutrients-12-01036-t003:** Predictors of resistance to insulin.

	Odds Ratio (95% CI)	*p*-Value
ND	1	
LD	0.724 (0.539–0.974)	0.033
MD	0.701 (0.577–0.852)	<0.001
HD	0.800 (0.686–0.933)	0.005

Explanatory variables include age, obesity, hypertension, fasting hyperglycemia, habit of drinking sweet beverages, and physical activity. CI, confidence interval.

**Table 4 nutrients-12-01036-t004:** Biological features of non-drinkers and various types of drinkers.

	ND	Mixed Drinkers	Beer Drinkers	Sake Drinkers	Shochu Drinkers	Whiskey Drinkers	Wine Drinkers	*p*-Value
N (%)	894 (36.8)	349 (14.4)	813 (33.5)	103 (4.2)	236 (9.7)	17 (0.7)	17 (0.7)	
Alcohol consumption (95% CI) (g/d)	0	54.3 ± 23.9 (51.8, 56.8)	20.4 ± 13.8 (19.4, 21.3)	32.9 ± 16.9 (29.6, 36.2)	43.7 ± 26.2 (40.3, 47.1)	33.4 ± 18.4 (23.9, 42.9)	16.4 ± 18.2 (7.0, 25.7)	<0.001
LD (%), MD (%), HD (%)	0	29 (8.3), 187 (53.6), 133 (38.1)	653 (80.3), 131 (16.1), 29 (3.6)	49 (47.6), 43 (41.7), 11 (10.7)	105 (44.5), 74 (31.4), 57 (24.2)	7 (41.2), 7 (41.2), 3 (17.6)	14 (82.4), 2 (11.8), 1 (5.9)	<0.001
Age (y)	53.4 ± 9.2	54.6 ± 8.3	53.2 ± 9.4	60.8 ± 8.3	56.9 ± 8.6	52.4 ± 9.2	56.8 ± 10.3	<0.001
BMI (kg/m^2^)	24.2 ± 3.4	24.2 ± 3.0	23.9 ± 3.1	23.4 ± 2.5	24.1 ± 3.2	24.0 ± 3.2	23.8 ± 2.8	0.243
AST (IU/L)	24 ± 9	26 ± 26 *	24 ± 9	27 ± 9	26 ± 9	22 ± 3	25 ± 8	0.004
ALT (IU/L)	30 ± 19	27 ± 22	28 ± 17	28 ± 15	27 ± 13	23 ± 8	24 ± 12	0.058
GGT (IU/L)	34 ± 30	66 ± 85 ***	47 ± 49 ***	59 ± 70 ***	67 ± 67 ***	42 ± 27	55 ± 51	<0.001
Triglycerides (mg/dL)	125 ± 86	142 ± 102	129 ± 80	121 ± 79	152 ± 149**	148 ± 114	119 ± 81	0.001
Use of lipid-lowering medications (%)	129 (14.4)	34 (9.7)	89 (10.9)	11 (10.7)	30 (12.7)	1 (5.9)	3 (17.9)	0.193
Fasting glucose (mg/dL)	103 ± 20	105 ± 20	104 ± 21	102 ± 14	107 ± 18	100 ± 10	102 ± 10	0.156
Use of antidiabetic medications (%)	62 (6.9)	18 (5.2)	43 (5.3)	3 (2.9)	24 (10.4)	0 (0)	1 (5.9)	0.061
HOMA-IR	1.9 ± 1.7	1.5 ± 1.1 **	1.6 ± 1.4 *	1.3 ± 0.9 *	1.5 ± 1.0*	1.7 ± 1.0	1.7 ± 2.0	<0.001
Fatty liver (%)	426 (47.7)	139 (39.8)	312 (38.4)	37 (35.9)	97 (41.1)	6 (35.3)	7 (41.2)	0.005

CI, confidence interval; BMI, body mass index; AST, aspartate aminotransferase; ALT, alanine aminotransferase; GGT, γ-glutamyl transferase; HOMA-IR, homeostasis model assessment-insulin resistance. * *p* < 0.05 vs. ND; ** *p* < 0.005 vs. ND; *** *p* < 0.001 vs. ND.

**Table 5 nutrients-12-01036-t005:** Prevalence of fatty liver in various types of drinkers.

	LD	MD	HD
Mixed drinkers (fatty liver/total number, (%))	13/29 (44.8)	76/187 (40.6)	50/133 (37.6)
Beer drinkers (fatty liver/total number, (%))	254/653 (38.9)	42/131 (32.1)	16/29 (55.2)
Sake drinkers (fatty liver/total number, (%))	14/49 (28.6)	15/43 (34.9)	8/11 (72.7)
Shochu drinkers (fatty liver/total number, (%))	41/105 (39.0)	32/74 (43.2)	24/57 (42.1)
Whiskey drinkers (fatty liver/total number, (%))	3/7 (42.9)	2/7 (28.6)	1/3 (33.3)
Wine drinkers (fatty liver/total number, (%))	4/14 (28.6)	2/2 (100)	1/1 (100)
*p*-Value	0.665	0.215	0.119

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
