# Peer review of "Chronic Alcohol Consumption is Inversely Associated with Insulin Resistance and Fatty Liver in Japanese Males"

_nutrients, 2020, doi:10.3390/nu12041036_

Round 1
Reviewer 1 Report
This is a retrospective, cross-sectional study performed in the male Japanese population aimed at assessing the association between NAFLD and the habitual alcohol consumption. Prevalence of NAFLD was lowest in moderate drinkers and highest in non-drinkers. Similar findings were observed for ALT elevation and for insulin resistance. Moderate alcohol consumption was the most protective for insulin resistance and for the prevalence of NAFLD.
A major limitation of the study is the fact that the study was conducted on a normal-weight Japanese population which is very different from that of western countries where fatty liver occurs in overweight or obese individuals with insulin resistance and often with diabetes and where the most important consumption of alcohol is represented by wine.
The study is interesting and well performed. However, there are some issues that need to be addressed/clarified to ensure that the data fully support the statements and conclusions made. I suggest that the Authors give some comments on the following points:
- The study population should be better described. Did the subjects undergo clinical examination consecutively? Where they outpatients or inpatients? Why the subjects participated in the physical check-up?
- It is necessary to add a table with the prevalence data of comorbidities and drug consumption of the study population.
- Describe alcohol consumption in mix drinkers.
- Prevalence of fatty liver and of HOMA-IR was higher in non-drinkers as compared to drinkers (table 4). Non-drinkers should be better characterized. Prevalence of comorbidities in drinkers and non-drinkers should be reported. For example, the prevalence of diabetes and of metabolic syndrome was the same in the two groups?
- 18 does not describe US diagnostic criteria for fatty liver.
- The title, the discussion and the conclusions are misleading. Since this is a cross-sectional study, the Authors cannot say that chronic consumption of alcohol improves resistance to insulin and protects against fatty liver. Rather, they can say that it is associated with insulin resistance and fatty liver.
- Please, better specify what the Authors mean by “apparently healthy subjects to engage in their work”
- The Authors should add as a major limitation of the study the fact that the study was conducted on the normal-weight Japanese population which is very different from that of western countries where fatty liver occurs in overweight or obese individuals with insulin resistance and often with diabetes and where the most important consumption of alcohol is represented by wine.
Author Response
Response to Reviewer 1:
We wish to express our appreciation to the reviewer for their insightful comments, which have helped us significantly improve the paper.
Comment 1: The study population should be better described. Did the subjects undergo clinical examination consecutively? Where they outpatients or inpatients? Why the subjects participated in the physical check-up?
Response: We have added additional information about the study population (line 49).
The subjects underwent annual examinations for employee or resident health check-ups.
Comment 2: It is necessary to add a table with the prevalence data of comorbidities and drug consumption of the study population.
Response: We have added the prevalence of hypertension and metabolic syndrome, and the use of drugs (antihypertensive, lipid lowering and antidiabetic medication) to Table 1, and have added the use of drugs (lipid lowering and antidiabetic medication) to Table 4.
We have also described it in the text, as seen below.
The prevalence of hypertension and the percentage of use of antihypertensive medications were shown to be significantly increased in association with the consumption of alcohol (lines 112-113)
The prevalence of metabolic syndrome was noted to be the lowest in the LD group and highest in the HD group. The percentages of the use of lipid lowering medications or antidiabetic medications were not significantly different among the 4 groups (lines 118-121).
Comment 3: Describe alcohol consumption in mix drinkers.
Response: We have described alcohol consumption and the percentage of the HD group in mix drinkers in the text. We have added the proportion of the categories of alcohol consumption in nondrinkers and various types of drinkers to Table 4.
Comment 4: Prevalence of fatty liver and of HOMA-IR was higher in nondrinkers as compared to drinkers (table 4). Nondrinkers should be better characterized. Prevalence of comorbidities in drinkers and nondrinkers should be reported. For example, the prevalence of diabetes and of metabolic syndrome was the same in the two groups?
Response: We have added the prevalence of metabolic syndrome to Table 1.
We have described this in the Result and Discussion sections, as seen below.
Result: The prevalence of metabolic syndrome was noted to be the lowest in the LD group and highest in the HD group (lines 118-119).
Discussion: The prevalence of metabolic syndrome was the highest in the HD group among the 4 categories of alcohol consumption. The reason was that triglycerides, and blood pressure, the risk factors for metabolic syndrome were the highest in the HD group among the 4 groups since alcohol intake is associated with triglyceride and the blood pressure in a dose-dependent manner (lines 237-241).
Comment 5: 18 does not describe US diagnostic criteria for fatty liver.
Response: We have deleted [18] because of mistyping.
Comment 6: The title, the discussion and the conclusions are misleading. Since this is a cross-sectional study, the Authors cannot say that chronic consumption of alcohol improves resistance to insulin and protects against fatty liver. Rather, they can say that it is associated with insulin resistance and fatty liver.
Response: The title has been changed to “Chronic Alcohol Consumption is inversely associated with Insulin Resistance and Fatty Liver in Japanese Males.”
Comment 7: Please, better specify what the Authors mean by “apparently healthy subjects to engage in their work”
Response: We have described this further in the text, as seen below.
Forth, most of the subjects were asymptomatic enough to engage in their work. The results of this study might thus not be applicable to individuals who have liver fibrosis or other liver diseases (lines 269-271).
Comment 8: The Authors should add as a major limitation of the study the fact that the study was conducted on the normal-weight Japanese population which is very different from that of western countries where fatty liver occurs in overweight or obese individuals with insulin resistance and often with diabetes and where the most important consumption of alcohol is represented by wine.
Response: We appreciate the reviewer's comment on this point. We have added it in the Discussion, as seen below.
Sixth, the study was conducted on the normal-weight Japanese population, which is very different from that of western countries where fatty liver occurs in overweight or obese individuals with insulin resistance and often with diabetes and where the most prevalent consumption of alcohol is represented by wine (lines 274-277).

Reviewer 2 Report
In this study by T. Akahane et al., authors describe the relationship between alcohol consumption and the development of fatty liver, concluding that moderated alcohol consumption may be protective against fatty liver and insulin resistance assessed by HOMA-IR. This is an interesting study, but present several, in the reviewer’s opinion, important limitation and flaws. These are listed below.
With regard to the assessment of alcohol consumption, what time frame before the assessment was used to evaluate the habitual consumption of alcohol? One year, 6 months, 1 month? This is a particularly relevant information to report. Although authors assessed the current alcohol, considering liver fat requires time to accumulate, and that current consumption may not necessarily match previous alcohol intake, it would be important to understand whether the current consumption is consistent with general alcohol consumption in, at least, the last 3 to 6 months. Have the authors access to this information?
Table 2: Multiple comparison should be performed. It looks like the authors only assessed the overall differences and not the difference between each group, this applies to table 1 and 4 and figure 1. It is important to note that while ALT does not increase with alcohol consumption, ASP and GGT seem to go up as the intake of alcohol increases, hence the liver is still affected by alcohol consumption. This should be further discussed.
A major concern in relation to the present study is that the authors did not make any reference to the diet of the study participants nor adjusted their results for dietary intakes. Particularly, consumption of specific nutrients such as fructose or saturated fatty acids have been shown to play a pivotal role in the development of fatty liver, nevertheless authors did not consider these variables at all. Assessing the BMI is not enough. Furthermore, do non-drinkers compensate for not consuming any alcohol by consuming other beverages (other than water) or foods? For example, they may drink more sugary soft drinks which in turn are rich in fructose, therefore it is particularly important to take dietary intakes into account to avoid potential confounders.
Minor point:
Line 30: please change “persons” to “people”
Line 48: please correct “cross-sectional”.
Author Response
Response to Reviewer 2:
We wish to express our appreciation to the reviewer for their insightful comments, which have helped us significantly improve the paper.
Comment 1: With regard to the assessment of alcohol consumption, what time frame before the assessment was used to evaluate the habitual consumption of alcohol? One year, 6 months, 1 month? This is a particularly relevant information to report. Although authors assessed the current alcohol, considering liver fat requires time to accumulate, and that current consumption may not necessarily match previous alcohol intake, it would be important to understand whether the current consumption is consistent with general alcohol consumption in, at least, the last 3 to 6 months. Have the authors access to this information?
Response: We appreciate the reviewer's comment on this point. We asked the participants about general alcohol consumption for the last year, not about current consumption.
We have described this in the text, as seen below.
Each subject reported their frequency of consumption of alcohol per week, as well as the types and volume of alcohol consumed per day for the last year (lines 85-87).
Comment 2: Multiple comparison should be performed. It looks like the authors only assessed the overall differences and not the difference between each group, this applies to table 1 and 4 and figure 1. It is important to note that while ALT does not increase with alcohol consumption, ASP and GGT seem to go up as the intake of alcohol increases, hence the liver is still affected by alcohol consumption. This should be further discussed.
Response: We thank the reviewer for this pertinent comment.
We have added the significance of individual differences for nondrinkers in Tables 1 and 4. We have added the data of the adjusted standardized residuals in the text and figure 1. The difference between nondrinkers and drinkers became clear by these analyses.
We have added explanations as to why the levels of ALT was a U-shaped association with the categories of alcohol consumption, and AST and GGT were increased with alcohol consumption in the Discussion, as seen below.
In the present study, GGT and AST were increased across the categories of alcohol consumption, whereas, there was a U-shaped association between the level of ALT and the categories of alcohol consumption. GGT, ALT, and AST are the most frequently used markers for early detection of alcoholic liver disease (ALD). GGT is usually higher in patients with ALD compared to those who have other liver diseases. Elevation of AST is observed in ALD, and ALT levels are commonly lower than AST. However, all these laboratory values are not specific for ALD, and they have low sensitivity and specificity. ALT has the lowest sensitivity among them [33]. The level of these values is also often elevated in NAFLD patients, and ALT levels are commonly higher than AST. Therefore, the elevation of ALT due to NAFLD in the ND group, elevation of ALT due to ALD in HD group and the lower prevalence of fatty liver in the LD and MD groups were observed, resulting in the U-shape relationship between the level of ALT and the categories of alcohol consumption (lines 242-253).
Comment 3: A major concern in relation to the present study is that the authors did not make any reference to the diet of the study participants nor adjusted their results for dietary intakes. Particularly, consumption of specific nutrients such as fructose or saturated fatty acids have been shown to play a pivotal role in the development of fatty liver, nevertheless authors did not consider these variables at all. Assessing the BMI is not enough. Furthermore, do nondrinkers compensate for not consuming any alcohol by consuming other beverages (other than water) or foods? For example, they may drink more sugary soft drinks which in turn are rich in fructose, therefore it is particularly important to take dietary intakes into account to avoid potential confounders.
Response: We have added information about dietary habits, such as the preference for soft drinks and greasy meat and physical activity for analysis. We have added this information to Table 1. Since the percentages of preference for soft drink and physical activity were significantly different among the categories of alcohol consumption, we have added them to the explanatory variables in the logistic regression analyses. The results were that chronic consumption of alcohol is independently associated with fatty liver and resistance to insulin after adjusting for obesity, hypertension, fasting hyperglycemia, preference for soft drinks and physical activity, and age.
Since we did not investigate nutrition, we have added this as a limitation of the study in the Discussion, as seen below.
Fifth, although the subjects answered dietary habits questions, such as soft drink and greasy meat, we did not investigate consumption of specific nutrition, such as fructose and saturated fatty acid, which have been shown to play a pivotal role in the development of fatty liver (lines 271-274)
Minor point:
Line 30: please change “persons” to “people”
Response: We have made this correction.
Line 48: please correct “cross-sectional”.
Response: We have corrected it.

Round 2
Reviewer 1 Report
I thank the Authors for for responding to all the raised comments.
Author Response
Response to Reviewer 1:
We wish to express our appreciation to the reviewer for their insightful comments, which have helped us significantly improve the paper.
The manuscript has been carefully reviewed by an experienced editor whose first language is English and who specializes in editing papers written by scientists whose native language is not English.

Reviewer 2 Report
Authors have addressed the reviewer’s comments. However, the paper, in the reviewer’s opinion, still requires revisions.
Authors report they did not “investigate nutrition” so “dietary habits” should be removed from the abstract as authors could not adjust for this component. The same should be done throughout the text. Indeed, the authors only refer to preference for greasy meat and soft drink. Does this directly reflect the consumption of these food items? How can the “preference” match the consumption of a food item? This is a point that still requires clarification from the authors. Adjusting for the “preference” of these food items is not sufficient to conclude that alcohol consumption is inversely and independently associated with insulin resistance and fatty liver especially considering that, by looking at soft drink preference and physical activity, MD may have a healthier lifestyle (lower preference for soft drink and higher-physical activity). Obviously, this is only an assumption considering the lack of data relative to dietary intakes. Also, especially in the absence of a nutritional assessment, stating that the consumption of alcohol is INDEPENDENTLY and inversely associated with fatty liver in misleading and not supported by the actual data. However, the reviewer agrees on the association after the adjustment for all the other variables considered as part of the study. Thus, considering the lack of nutritional data, the word independently should be removed from the text when referring to this association and emphasise the fact that data are only corrected for some dietary preferences and not actual dietary intakes. Despite the authors having described the limitation of not assessing nutritional parameters, this remains a crucial limitation of the study and should be further reiterated and made clear throughout the text including the abstract.
Finally, in table 1 the difference between groups for “Prefer soft drink”, “Physical activity (%)”, “Hypertension (%)”, “Use of antihypertensive medications”, “Metabolic syndrome (%)” should be reported.
Line 204: It appears that there is an “inversely” missing. Please double check.
Author Response
Response to Reviewer 2:
We wish to express our appreciation to the reviewer for their insightful comments, which have helped us significantly improve the paper.
Comment 1: Authors report they did not “investigate nutrition” so “dietary habits” should be removed from the abstract as authors could not adjust for this component. The same should be done throughout the text. Indeed, the authors only refer to preference for greasy meat and soft drink. Does this directly reflect the consumption of these food items? How can the “preference” match the consumption of a food item? This is a point that still requires clarification from the authors. Adjusting for the “preference” of these food items is not sufficient to conclude that alcohol consumption is inversely and independently associated with insulin resistance and fatty liver especially considering that, by looking at soft drink preference and physical activity, MD may have a healthier lifestyle (lower preference for soft drink and higher-physical activity). Obviously, this is only an assumption considering the lack of data relative to dietary intakes. Also, especially in the absence of a nutritional assessment, stating that the consumption of alcohol is INDEPENDENTLY and inversely associated with fatty liver in misleading and not supported by the actual data. However, the reviewer agrees on the association after the adjustment for all the other variables considered as part of the study. Thus, considering the lack of nutritional data, the word independently should be removed from the text when referring to this association and emphasise the fact that data are only corrected for some dietary preferences and not actual dietary intakes. Despite the authors having described the limitation of not assessing nutritional parameters, this remains a crucial limitation of the study and should be further reiterated and made clear throughout the text including the abstract.
Response: We thank the reviewer for this pertinent comment.
We deleted “dietary habits” in the abstract and the text. (Lines 21, 58).
We deleted “independently” in the conclusions. (Lines 278-279).
We described the limitation of not assessing diet or nutrition in the abstract and the text. (Line 24-25, 270-273)
Comment 2: Finally, in table 1 the difference between groups for “Prefer soft drink”, “Physical activity (%)”, “Hypertension (%)”, “Use of antihypertensive medications”, “Metabolic syndrome (%)” should be reported.
Response: We have added the data of the adjusted standardized residuals in the text and table 1. (Lines 104-105, 107-109)
Minor point:
Line 204: It appears that there is an “inversely” missing. Please double check.
Response: We have inserted “inversely”.
The manuscript has been carefully reviewed by an experienced editor whose first language is English and who specializes in editing papers written by scientists whose native language is not English.
